# The Role of Gut Dysbiosis in Acute-on-Chronic Liver Failure

**DOI:** 10.3390/ijms222111680

**Published:** 2021-10-28

**Authors:** Sung-Eun Kim, Ji Won Park, Hyung Su Kim, Myoung-Kuk Jang, Ki Tae Suk, Dong Joon Kim

**Affiliations:** 1Department of Internal Medicine, Hallym University College of Medicine, Chuncheon 24252, Korea; sekim@hallym.or.kr (S.-E.K.); miunorijw@hallym.or.kr (J.W.P.); hskim@kdh.or.kr (H.S.K.); mkjang@kdh.or.kr (M.-K.J.); ktsuk@hallym.ac.kr (K.T.S.); 2Institute for Liver and Digestive Diseases, Hallym University, Chuncheon 24252, Korea

**Keywords:** cirrhosis, gut microbiome, alcoholic liver disease, acute-on-chronic liver failure

## Abstract

Acute-on-chronic liver failure (ACLF) is an important syndrome of liver failure that has a high risk of short-term mortality in patients with chronic liver disease. The development of ACLF is associated with proinflammatory precipitating events, such as infection, alcoholic hepatitis, and intense systemic inflammation. Recently, the role of the gut microbiome has increasingly emerged in human health and disease. Additionally, the gut microbiome might have a major role in the development of liver disease. In this review, we examine evidence to support the role of gut dysbiosis in cirrhosis and ACLF. Additionally, we explore the mechanism by which the gut microbiome contributes to the development of ACLF, with a focus on alcohol-induced liver disease.

## 1. Introduction

The gut microbiome is a complex ecosystem consisting of more than 100 trillion microorganisms [1]. In adulthood, the gut microbiome tends to show rather stable health, but it is affected by age, pharmaceuticals, diet, alcohol, and smoking [2,3]. The physiological interaction between the gut microbiome and host has an important role in metabolic functions, such as production of vitamins K and B and fatty acids and glucose metabolism, and it also contributes to the immune system of the host, with various critical roles revealed [4,5]. Gut dysbiosis, which is defined as imbalances in the composition of the gut microbiome, has been associated with many diseases, such as gastrointestinal, hepatic, cardiovascular, respiratory, metabolic, neurological, and psychiatric diseases, including malignancies [5].

The concept of the gut–liver axis is increasingly emphasized in liver, gastrointestinal, and immune diseases [6,7]. The liver communicates with the gut through the biliary tract, portal vein, and systemic circulation. Endogenous (bile acids and amino acids) and exogenous (diet and drugs) substrates metabolized by the host and gut microbiome directly enter the liver via the portal circulation after crossing the epithelial barrier of the gut [8]. Intestinal permeability is increased by gut inflammation and gut dysbiosis [9,10], which have been associated with a high-fat diet [11,12], chronic alcohol consumption [13,14,15], long-term use of antibiotics [16], and inflammatory bowel disease [17]. Breakdown of the intestinal epithelial barrier allows bacteria or their metabolites to enter the liver through the portal system, causing hepatic damage and inflammation [5].

Globally, acute and chronic liver disease (CLD), including cirrhosis and hepatocellular carcinoma, account for approximately 2 million deaths per year [18,19]. The causes of CLD vary. In North America and Europe, alcohol-related liver disease (ALD), nonalcoholic fatty liver disease (NAFLD), and hepatitis C virus (HCV) infection are more common [19,20], whereas in Asia, hepatitis B virus (HBV) infection is the predominant etiology [21]. Regardless of the etiology, CLD can progress to liver fibrosis and finally cirrhosis if the cause is not eliminated [22]. Cirrhosis is characterized by diffuse nodular regeneration surrounded by fibrous septa and severe disruption of the intrahepatic arterial and venous flow and portal hypertension. Clinically, cirrhosis is classified as compensated or decompensated. Progression to decompensated cirrhosis is associated with 3- to 5-year survival, and patients with decompensated cirrhosis are warned to prepare for liver transplantation [23].

Acute decompensation of cirrhosis and acute-on-chronic liver failure (ACLF) are two major challenges in patients with CLD. Acute decompensation of cirrhosis is defined as the development of ascites, hepatic encephalopathy, jaundice, and/or gastrointestinal hemorrhage only in cirrhotic patients [24,25]. ACLF is characterized by failure of one or more major organs (liver, kidney, brain, coagulation, circulation, or respiration). Patients who develop ACLF have high short-term mortality (within 28 days after admission) [26]. The definition of ACLF differs depending on the region of the world. The European Association for the Study of the Liver-Chronic Liver Failure Consortium and the North American Consortium for the Study of End-Stage Liver Disease (NACSELD) define the concept of ACLF only in cirrhotic patients [24,27], whereas the definition of the Asian Pacific Association for the Study of the Liver (APASL) does not require cirrhosis to define ACLF. Furthermore, unlike the Western definition of ACLF, the APASL includes only liver failure (jaundice and coagulopathy) [25]. In Asia, HBV infection (76%) is a major cause of ACLF [25,28,29], which is associated with the increased development of liver and coagulation failure. As the case stands, the APASL definition does not require the status of cirrhosis. However, the Korean Acute-on-Chronic Liver Failure study involving 1470 prospectively enrolled cirrhotic patients demonstrated that the major etiology was alcohol (72%) [28]. Additionally, ACLF significantly occurs among those with alcoholic cirrhosis (60%) and is caused by infection, alcohol, or both in Western countries [24].

The pathophysiology of ACLF has largely been studied but is still unknown. Many studies have reported that systemic inflammation from bacterial infection and alcohol directly correlate with the severity of ACLF [29,30,31]. Approximately 40–50% of patients with ACLF have systemic inflammation without any identifiable precipitating triggers [24]. The mechanism of systemic inflammation suggests that metabolites produced by the gut microbiome may affect the systemic compartment and trigger systemic inflammation [32]. Systemic inflammation may induce single or multiple organ failure in patients with cirrhosis. Therefore, the role of gut dysbiosis could be considered an important factor in managing the precipitating factor, diagnosis, treatment, and prevention of ACLF [33]. This review discusses the role of dysbiosis in patients with ACLF and highlights the role of the gut microbiome in patients with alcohol-induced ACLF.

## 2. Gut Dysbiosis in Cirrhosis

Cirrhotic patients have increased translocation of intestinal bacteria and circulating concentrations of bacterial DNA. Additionally, the weakened gut epithelial barriers can easily allow translocated bacteria and their metabolites to influx into the portal blood circulation. The translocated bacteria interact with liver cells, some of which alter the gut microbiome composition and act as signaling molecules (Table 1) [34].

### 2.1. The Profile the of Gut Microbiome in Cirrhosis

The interaction between the gut microbiome and liver is highly dependent on the etiology of cirrhosis, despite a lack of sufficient data [36,42]. Chen et al. demonstrated that the fecal microbial communities in cirrhotic patients (*n* = 36; 24 HBV-related and 12 alcohol-related) compared with healthy subjects (*n* = 24) were analyzed by 454 pyrosequencing of the 16S ribosomal RNA V3 region and by subsequent real-time PCR. At the phylum level, Bacteroidetes and Firmicutes were prevalent in the fecal microbial communities of both cirrhotic patients and healthy subjects. However, this study revealed that the proportion of the phylum Bacteroidetes was significantly decreased, whereas Proteobacteria and Fusobacteria were significantly increased in cirrhotic patients. At the family level, Enterobacteriaceae, Veillonellaceae, and Streptococcaceae were prevalent in cirrhotic patients. The Child-Turcotte-Pugh (CTP) score was positively correlated with Streptococcaceae, whereas the proportion of Lachnospiraceae was reduced significantly in cirrhotic patients and correlated negatively with the CTP score [35]. Interestingly, although no significant difference was observed at the phylum and class levels of the gut microbiome between HBV-related and alcohol-related cirrhosis, they reported a profound increase in Prevotellaceae in alcohol-related cirrhosis [35]. Additionally, a prior study reported that Lachnospiraceae was significantly decreased in the colonic mucosa of cirrhotic patients as compared with that of healthy subjects [40]. The author demonstrated that Virbionaceae, Enterobacteriaceae, and Alcaligenaceae were prevalent in the colonic mucosa of cirrhotic patients as compared with those of healthy subjects [40]. Among cirrhotic patients, there was a difference in the composition of gut microbiota between alcohol-related patients and NAFLD-related patients, which are described in Table 1 [36]. Interestingly, the model for end-stage liver disease (MELD) score was negatively correlated with Clostridiales XIV, Lachnospiraceae, Ruminococcaceae, and Rikenellaceae but positively correlated with Staphylococcae, Enterococceae, and Enterobacteriaceae. There was also a significant association of the cirrhosis dysbiosis ratio (CDR) with the MELD score and endotoxin [36].

Lachnospiraceae has a role in carbohydrate fermentation into short-chain fatty acids (SCFAs) and CO_2_ with H_2_ in the human intestine [43]. Lachnospiraceae and Ruminococcaceae, with butyrate production roles, were enriched in healthy subjects. A lower proportion of these species in cirrhotic patients indicates that cirrhotic patients have a less healthy gut microbiome [38]. The reduction in these fermentation-related gut microbiomes induces a decline in SCFA production. The role of SCFAs includes their role as nutrients for the colonic epithelium as modulators of colonic pH [44]. Surprisingly, taxa such as *Veillonella* or *Streptococcus*, known as species of oral origin, were enriched in the small intestines of cirrhotic patients [45]. These oral commensals can invade the gut in cirrhotic patients. Additionally, small-intestinal bacterial overgrowth is easily found in cirrhotic patients [41].

### 2.2. Progression to Decompensated Cirrhosis

During liver disease progression, host mucosal proteins and pathways, such as the farnesoid X receptor (FXR) signaling pathway, are changed by an altered gut microbiome and its metabolites, such as SCFAs [46]. Additionally, the alteration of gut innervations might influence the gut epithelial barrier, induce gut inflammation [47,48], and reduce the levels of antibacterial peptides [49]. Gut-associated lymphatic tissues might affect gut epithelial barrier dysfunction, thereby increasing the permeability of the gut epithelial barrier [46,50]. Liver disease progression is related to systemic inflammation, leading to impaired dendritic cell activity, increased tumor necrosis factor (TNF)- and interferon (IFN)-γ-expressing lymphocytes, and depletion of interleukin (IL)-17-producing T helper cells [46]. Patients with cirrhosis show increased levels of lipopolysaccharide (LPS) and bacterial DNA in the portal blood circulation, as compared with those in healthy subjects, and these levels become higher as liver function worsens [51,52]. Schierwagen et al. [53] reported that bacteria from the circulating blood microbiome and their compartment-specific patterns are viable in patients with transjugular intrahepatic portosystemic shunts. This suggests that liver bacteria enable translocation across the gut epithelial barrier in patients with decompensated cirrhosis.

Gut dysbiosis and bacterial translocation are significantly associated with the development of acute decompensation of cirrhosis and ACLF [38,54,55]. In fact, the treatment of variceal bleeding episodes with antibiotics or long-term prophylactic use of antibiotics in decompensated cirrhosis improves outcomes [54]. Therefore, gut dysbiosis in cirrhotic patients may be the main factor for disease progression.

Interestingly, Bajaj et al. also reported that alterations in the gut microbiome are associated with the severity and stability of cirrhosis over time. This study demonstrated that the ratio of autochthonous to nonautochthonous taxa was measured as the CDR using multitagged pyrosequencing. In the study, the gut microbiome profile in patients with cirrhotic changes worsened in patients with poor outcomes but remained stable in patients with a stable status [36]. Therefore, the interaction of the gut microbiome and liver cirrhosis per se is the main factor of disease progression (Table 1).

### 2.3. Gut Dysbiosis and Gut-Brain Axis in Hepatic Encephalopathy

Several studies have clearly revealed the bidirectional communication between the gut microbiota and the brain [55]. The gut microbiota can send signals to the brain via neuronal-, endocrine-, and immune-mediated pathways [56]. The amount and type of the signals from gut microbiota to the brain are highly dependent on the regional intestinal environment because the gut microbiota is influenced by various factors such as intestinal transit, mucus secretion, antimicrobial peptides, and intraluminal release of neuroactive substances, intestinal permeability, blood-brain barrier, and the clearance of gut microbial metabolites [55,57]. Hepatic encephalopathy (HE) in cirrhotic patients shows a very wide spectrum, ranging from minimal HE, including reversal of sleep-wake cycle, short-term memory loss, poor concentration, and deficits that may only become apparent on neurocognitive test, to the most severe symptoms of coma [58]. HE is a major gut dysbiosis-related complication in cirrhotic patients. This is a result of systemic endotoxemia and inflammation that ultimately induces neuroinflammation, although ammonia was noted to be central to the pathogenesis of HE [59,60,61]. Interestingly, Bajaj et al. demonstrated that the IL-23 system was significantly associated with several bacterial families in patients with HE, and there was a direct correlation between cognition, *Porphyromonadaceae*, and *Alcaligeneceae* [38]. Additionally, Ahluwalia et al. [62] showed that patients with HE had more evidence of systemic inflammation, gut dysbiosis, and hyperammonemia compared with healthy subjects and cirrhotic patients without HE. Several studies demonstrated that the altered linkage between gut microbiota and the brain had a key role in the development of HE. The findings of small intestinal overgrowth, gut dysbiosis, increased intestinal permeability, endotoxemia, and changes in brain status are consistent with this concept. Thus, altered brain–gut microbiome communication might provide novel therapeutic approaches and the prediction of outcome in patients with HE.

## 3. Gut Dysbiosis and Progression to ACLF

ACLF induced by acute insult or pathogens directly or indirectly activates different cells and inflammatory cytokines. Danger-associated molecular patterns (DAMPs) and other cytokines are released from injured parenchymal and nonparechymal cells [61]. Liver tissue damage is induced by the direct action of virulence factors, excessive immune response, and failure of the immune-mediated tolerance system of the host [63]. The deterioration from decompensated cirrhosis to ACLF is associated with extensive systemic inflammation activating many inflammatory systems and cytokine pathways [29,63]. The association between ACLF and DAMPs is particularly important in patients with ACLF due to viral activation, such as HBV reactivation [64,65], as well as superimposed hepatitis A virus and hepatitis E virus infections [65,66]. This systemic inflammation and organ failure caused by bacterial infection occurs in approximately 30% of patients with ACLF [24]. The most common cause in these patients is spontaneous bacterial infection [24]. As a result of infection, pathogen-associated molecular patterns (PAMPs) release and stimulate intracellular signaling cascades, such as nuclear factor-κB. Then, activated transcription factors induce various gene-encoding molecules, such as cytokines and chemokines [67,68,69]. Elevated serum levels of IL-8 or IL-6 with or without obvious bacterial infection were demonstrated in patients with acute decompensation of cirrhosis and ACLF [29,63,70]. This understanding was also shown in the PREDICT (PREDICTing Acute-on-Chronic Liver Failure) study; patients in the pre-ACLF group showed significantly higher systemic inflammation than patients with acute decompensation [71]. Additionally, several studies have revealed that gut dysbiosis is a strong factor for the development of ACLF [33,70]. In a prospective study across North America, stool gut microbiome composition on admission was associated with the development of ACLF, individual organ failure, and mortality. This study showed that the phylum Proteobacteria was higher in patients with poor outcomes [72]. This result is not surprising because Proteobacteria, such as *Escherichia coli, Klebsiella*
*pneumoniae**,* and *Pseudomonas aeruginosa*, are responsible for infections in patients with cirrhosis [32,73,74,75,76]. However, increased pathogenic taxa, such as Enterococcaceae and Streptococcaceae, were highly associated with mortality [30,72,73]. Decreased autochthonous taxa, such as Lachnospiraceae and Clostridia, which are associated with the formation of secondary bile acids and SCFAs [74,75], were demonstrated [71]. Another study in patients with HBV-associated ACLF revealed that increased Prevotellaceae was an important risk factor for 28-day mortality, although the abundances of Enterobacteriaceae, Proteobacteria, and Fusobacteria were not significantly changed in the circulation of either the cirrhosis or HBV-associated ACLF group [74]. Zhang et al. demonstrated that increased Pasteurellaceae was an important risk factor for short-term mortality in patients with ACLF, mostly composed of HBV-associated ACLF. In this study, patients with ACLF had higher levels of inflammatory cytokines (TNF-α, IL-6, and IL-2); increased abundances of Fusobacteriaceae, Veillonellaceae, and Enterobacteriaceae; and a reduction in Ruminococcaceae [77]. Additionally, the negative association of Ruminococcaceae and Lachnospiraceae with TNF-α and IL-6 could anticipate a therapeutic role in ACLF [77]. Another recent metagenomics study confirmed that ACLF was associated with an increased abundance of *Enterococcus* and *Peptostreptococcus*. Interestingly, this study showed that cirrhotic patients had enriched pathways related to ethanol production, γ-aminobutyric acid metabolism, and endotoxin synthesis [78]. Alcohol-induced ACLF is the second highest after infection-induced ACLF, and several studies on gut dysbiosis related to alcohol consumption have been reported [24,36,78]. Associated studies between gut dysbiosis and alcohol-induced liver disease, including ACLF, will be described in more detail in the next section. Recently, studies on drug-induced ACLF [79] and autoimmune hepatitis-related ACLF [80,81] have been reported, but studies on gut dysbiosis in these patients are also warranted. Although many studies have noted that systemic inflammation and single or multiple organ failure in patients with ACLF are caused by gut dysbiosis and altered metabolic pathways, as well as by many of the altered metabolites from microbial dysbiosis, there is an urgent need to investigate whether there is a difference in the gut microbiome in cirrhosis with regard to the etiology and whether there is a change in the development pattern of ACLF and the prediction, treatment, and prevention of ACLF using the gut microbiome (Table 2).

### 3.1. Infection-Induced ACLF and Gut Dysbiosis

Bacterial infection is more common in cirrhotic patients than in healthy subjects [82]. Proteobacteria is highly associated with poor outcomes in patients with ACLF, whereas Firmicutes is associated with relatively good outcomes in patients with ACLF [36,77]. Recently, one multicenter clinical study demonstrated that alpha and beta diversity, as well as the CDR, were not different at admission between patients who did or did not develop outcomes, including the development of ACLF and mortality [69]. This study demonstrated that patients with ACLF who had infection on admission had higher abundances of Enterococcaceae and Staphylococcaceae and a lower abundance of Bifidobacteriaceae [69]. Changes in the Proteobacteria family, such as *E. coli*, *K. pneumoniae*, and *P. aeruginosa*, are responsible for infections, which lead to ACLF and death [30,69,72,83]. Additionally, the enrichment of Gram-positive pathogenic taxa, such as Enterococcaceae and Streptococcaceae, was highly associated with patient mortality [30,69,72,73]. These patients had a reduction in autochthonous taxa, such as Clostridia and Lachnospiraceaeae [69]. Bacterial translocation with gut dysbiosis is responsible for infection-induced ACLF. Proactive plans to reduce gut dysbiosis (and subsequent development of ACLF) in high-risk patients need to be studied to achieve favorable outcomes. Additionally, further studies are warranted with a larger number of patients with ACLF to clarify the independent role of the gut microbiome in the treatment and prediction of prognosis. 

### 3.2. Extrahepatic Organ Failure in ACLF and Gut Dysbiosis

In the CANONIC study, renal failure was the most common organ failure (55.8%), followed by hepatic failure (43.6%), coagulation failure (27.7%), brain failure (24.1%), circulation failure (16.8%), and respiratory failure (9.2%) [24]. The NACSELD study demonstrated that grade III–IV hepatic encephalopathy was the most common organ failure (55.7%), followed by circulatory failure (17.6%), respiratory failure (15.8%), and renal failure (15.1%) [84]. Although gut dysbiosis related to ACLF has been reported, Bajaj et al. investigated the difference in gut microbiome composition according to individual organ failure. The author reported that the most common extrahepatic organ failure was the brain (13%), followed by the kidneys (8%), lungs (7%), and heart (7%) [69]. The proinflammatory milieu is linked to the enrichment of Proteobacteria and the reduction in Firmicutes [36,69]. These gut microbiota are associated with endotoxin production, leaky gut barrier, and systemic inflammation that affects extraintestinal organs [30]. Proteobacteria (Xanthomonadaceae and Enterobacteriaceae), Tenericutes (Anaeroplasmataceae), Firmicutes (Erysipelotrichaceae), and Actinobacteria (Bifidobacteriaceae) were higher in those who developed brain failure, whereas Fusobacteriaceae were higher in those who remained free of it [69]. HE is an important complication of cirrhosis related to the gut microbiome and its byproducts, such as ammonia and endotoxin, in the leaky gut barrier and systemic inflammation [85,86,87,88]. Interestingly, a study of the change of gut dysbiosis after liver transplantation showed delta Proteobacteria and delta Firmicutes instead of Proteobacteria and Firmicutes before liver transplantation, which demonstrated that delta Proteobacteria was associated with cognitive improvement [89]. Alcaligeneceae and Prophyromonadaceae were linked to poor cognition in cirrhotic patients [37]. Alcaligeneceae were associated with opportunistic infections and the degradation of urea to produce ammonia [90]. *Porphyromonas* were related to both reduced stomach acid and bile barrier function in cirrhotic patients [37]. Renal failure was associated with MELD, infection on admission, and a higher abundance of Hydrogenophilaceae [69]. In addition, renal dysfunction was linked to a higher abundance of Enterobacteriaceae and lower abundance of Lachnospiraceae [39]. Circulatory failure was predicted by diabetes, admission infections, and a higher relative abundance of Enterobacteriaceae. Respiratory failure was related to a higher MELD score and the enrichment of Streptococcaceae [69]. Further studies are needed to clarify the independent role of the gut microbiome according to the failure of affected organs in ACLF.

### 3.3. Intervention on Gut Dysbiosis in ACLF

Restoration of gut dysbiosis is a major target in treatments for liver cirrhosis, including ACLF. Especially, fecal microbiota transplantation (FMT) demonstrated sustainable changes in commensal diversity with improvement in cognitive function and reduction in hospitalization in patients with HE [91]. New application of FMT using oral capsule demonstrated a similar therapeutic effect and safety profiles in cirrhotic patients with HE [92]. However, DeFilipp et al. reported that one patient died from extended-spectrum beta-lactamase-producing *Escherichia coli* bacteremia after he had undergone FMT [93]. Therefore, we clear definition of the benefits and risks of FMT across different patients are warranted.

Rifaximin, an oral non-absorbable antibiotic, is effective in treating recurrent HE [94]. In addition, a recent study showed improvement in the myriad complications of acute decompensation in cirrhotic patients [95]. Interestingly, Vlachogiannakos et al. reported that long-term rifaximin use reduced the risk of developing the complication of portal hypertension, and it improved survival [96]. However, other studies did not demonstrate the effect on progression of liver disease and the significant changes in the gut dysbiosis [97,98]. Quinolones have been used as prophylaxis in patients deemed to be at risk of spontaneous bacterial peritonitis. This primary prophylactic use of non-absorbable antibiotics has been shown to improve survival [99]. The advantages and duration of antibiotic prophylaxis need to consider concerns about the development of multidrug-resistant infections [100]. 

Probiotics have been shown to have a beneficial effect on cirrhotic patients with HE and are associated with a reduction in infection rates and ammonia level without survival improvement [101].

## 4. Gut Dysbiosis and Alcohol in ACLF

Alcoholic hepatitis (AH) is a major precipitating factor of ACLF, representing approximately 24.5% of the cases of ACLF [24]. Before CLD develops, alcohol consumption per se induces significant gut dysbiosis with a weakened gut barrier [102]. Increased gut permeability in alcoholics is shown in patients with altered composition and activity of the gut microbiome, such as *Bifidobacterium*, *Clostridium XIV*, *Incertae Sedis*, and Ruminococcaceae, as compared with those in healthy subjects [102]. Interestingly, when the gut microbiome of heavy drinking subjects with severe AH are transplanted into germ-free mice and fed an ethanol-containing diet, severe hepatic inflammation and weakened intestinal permeability are induced in the mice [103]. Short-term treatment using *Bifidobacterium* and *Lactobacillus* was related to restoration of the normal gut microbiome in ALD [104]. After transplantation of the gut microbiome in healthy subjects, liver injury was ameliorated, despite ongoing alcohol consumption [103].

With prolonged and harmful alcohol consumption, microbial diversity and enrichment of pathogenic bacteria, such as Enterobacteriaceae and Enterococcaceae, are further reduced. Gut dysbiosis in ALD is relatively more important than other etiologies because alcohol has direct toxicity to both the gut barrier and the gut microbiome before the onset of CLD [76,103,105]. Bacterial endotoxins, such as LPS secreted by Gram-negative bacteria; exotoxins such as candidalysin; and PAMPs from all types of gut microbiomes can induce liver injury. Endotoxins bind hepatic toll-like receptors, and PAMPs bind to pattern-recognition receptors on hepatic stellate cells and Kupffer cells. Then, these microbial products can promote an inflammatory cascade of cytokine release, oxidative stress, and fibrotic processes [106]. Exotoxins from cytolysin-producing *Enterococcus faecalis* were increased in patients with AH as compared with those in heavy drinking subjects without hepatitis, with the amount of cytolysin associated with disease severity and mortality, although the amount of fecal *Enterococcus faecalis* did not correlate with disease severity or mortality in patients with AH [107]. Candidalysin secreted by *Candida albicans* induces direct liver injury and exacerbates ethanol-induced liver injury in mice. Additionally, fecal positivity for candidalysin is related to disease severity and mortality in patients with AH [108]. The association of endotoxin and gut dysbiosis in ALD has been demonstrated in several studies [36,109,110,111]. Despite similar MELD scores, alcohol-induced cirrhotic patients have higher levels of endotoxin than nonalcohol-induced cirrhotic patients [36]. Intestinal permeability was highly associated with gut dysbiosis-linked endotoxemia. In addition, *Akkermansia muciniphila*, which promotes mucus thickening and gut barrier function, constitutes 1–4% of the fecal microbiota. This gut microbiome did not have the ability to metabolize ethanol, but it was protective against the disruption of the gut barrier induced by ethanol [112]. Therefore, gut dysbiosis seems to have a major role in changes in gut permeability and disease progression [102,113].

In general, most of the primary bile acids secreted into the gut are reabsorbed back into the portal circulation, whereas only 5% of primary bile acids are changed to secondary bile acids by the gut microbiome. Therefore, gut dysbiosis can alter bile acid metabolism, aggravate secondary bile acid conversion and reduce the rate of primary bile acid reabsorption [106]. Bile acids and the gut microbiome interact and modulate each other closely through bile acids binding FXR, which results in the production of antimicrobial peptides, such as angiogenin 1 and RNAse family member 4. Then, it induces the inhibition of gut microbial overgrowth and gut barrier function [114,115]. In patients with liver disease with ongoing active alcohol drinking, several studies have demonstrated a significant increase in secondary bile acids [103,116,117]. Fibroblast growth factor (FGF)-19 was induced through bile acids binding to FXR. Then, FGF-19 modulated the downregulation of de novo bile acid synthesis by inhibiting CYP7A1 in hepatocytes using a feedback system for modulation of bile acid production [118]. Interestingly, patients with AH showed higher elevations of both hepatic and circulating FGF-19 expression, although patients with nonalcoholic steatohepatitis did not show these findings. Additionally, the level of FGF-19 expression was associated with the MELD score in patients with AH [119].

Butyrate and propionate, as products of bacterial fermentation, are major SCFAs in the gut. Butyrate has a role in maintaining the gut barrier and providing an energy resource for enterocytes [120]. Chronic alcohol consumption results in gut dysbiosis characterized by a reduction in the SCFA-producing gut microbiome, such as Lachnospiraceae and Ruminococcaceae [35,110,121,122,123,124].

ALD is a broad spectrum of diseases ranging from asymptomatic liver steatosis to the progression of hepatitis, fibrosis, cirrhosis, and ACLF [125]. AH is a specific entity and can coexist with any stage of liver disease, even cirrhosis [126]. This section will focus on the gut dysbiosis associated with alcohol-related ACLF (Table 3).

### 4.1. Gut Dysbiosis and Gut–Brain Axis in ALD

The gut-brain axis is the connection between the gut and the brain by various metabolites, neural connections, and hormones. It is an important concept for understanding patients with alcohol use disorders (AUD), especially patients with ALD. The brain is affected by alcohol ranging from acute intoxication to changes in personality and behavior [128]. In patients with cirrhosis or ACLF, the brain reserve and functioning can easily worsen in patients with AUDs [129,130]. Leclercq et al. demonstrated that depression, anxiety, and alcohol craving was associated with increased intestinal permeability. This study showed that patients with low intestinal permeability recovered completely at the end of a 3-week detoxification program for depression and anxiety. Conversely, patients with high intestinal permeability were persistent in having depression, anxiety, and alcohol craving, even after alcohol withdrawal [102]. As a result of gut dysbiosis in patients with AUD, systemic inflammatory mediators, ammonia, and endotoxemia lead to worsened neuroinflammation [131]. In addition, the altered gut-brain axis can affect concomitant eating disorders, cocaine use, and anxiety disorders [131,132]. Additionally, the direct effect of alcohol on the brain or nutritional deficiencies in patients with AUD induces worsened brain function [133]. Recently, the manipulation of the gut-brain axis in patients with AUD, alcohol cravings, alcohol consumption, and long-term AUD-related hospitalizations was shown to potentially decrease after fecal microbiota transplantation, but not in the placebo group [134]. Given the role of the gut microbiota and altered intestinal permeability in patients with AUD, this approach is another promising area that requires dedicated investigation.

### 4.2. Gut Dysbiosis in Mild ALD

Gut dysbiosis was described via analysis of jejunal aspirates in heavy alcoholics in 1984. There were no significant differences between different stages of ALD [135]. With advances in stool analysis by PCR fingerprinting, Mutle et al. [109] reported a stepwise decrease in Bacteroidetes in healthy subjects, heavy drinkers without liver disease, and those with ALD. None of them showed gut dysbiosis; however, the gut microbiome was characterized by decreases in Bacteroidetes and increases in Proteobacteria [109]. A recent study revealed that patients with alcohol dependence syndrome had significant increases in *Klebsiella pneumonia, Lactobacillus salivarius, Citrobacter koseri,* and *Lactococcus lactis* subsp. *cremoris* and decreases in *Akkermansia, Coprococcus*, and unclassified Clostridiales as compared with those in healthy subjects. Interestingly, both alcohol dependence and alcoholic cirrhosis have enrichment of species belonging to the *Lactobacillus* and *Bifidobacterium* genera, although this study did not show significant changes in microbial diversity between any of the patient groups [121].

### 4.3. Gut Dysbiosis in Alcoholic Cirrhosis

Gut dysbiosis in patients with cirrhosis was largely irrespective of the etiology. Provotellaceae was enriched in patients with alcoholic cirrhosis compared with that in patients with HBV-related cirrhosis [35]. The reason was speculated that the enrichment of Provotellaceae in alcoholic cirrhosis may be associated with ethanol metabolism in the gut. Extrahepatic ethanol removal constitutes approximately 40% of total ethanol removal [136]. Microbial oxidation in the gut has an important role in extrahepatic ethanol removal [137]. The gut microbiome profile of patients with alcoholic cirrhosis had relatively higher abundances of Lachnospiraceae and Ruminococcacea [36]. Dubinkina et al. [121] noted increased abundances of *Bifidobacterium, Sterptococcus*, and *Lactobacillus* in patients with alcoholic cirrhosis, as well as decreased abundances of *Paraprevotella*, *Alistipes*, and *Prevotella*, as compared with those in healthy subjects. As previously mentioned, both *Lactobacillus* and *Bifidobacterium* are beneficial gut microbiomes with promising roles as probiotic supplements. Especially in alcoholic cirrhosis, the oral microbiome is important because these patients probably have high rates of periodontitis, changes in the oral microbiome, proton pump inhibitor use, and relatively low gastric acid levels [37,121,138,139,140]. For the purpose of evaluating the effects of continued drinking on the gut microbiome, Bajaj et al. obtained stool samples and mucosal biopsies taken at various sites in the gut of patients who were either actively drinking or not actively drinking. They reported that Lachnnospiraceae, Ruminococcaceae, and Clostridiales cluster XIV were significantly decreased in stool samples and all mucosal samples of actively drinking cirrhotic patients as compared with those in cirrhotic patients with abstinence and with controls [116]. Enterobactericaceae, *Enterobacter,* and *Bacteroides* were more enriched in alcoholic cirrhosis, whereas the proportion of *Bifidobacterium* and *Lactobacillus* was not different between different groups (alcoholics with cirrhosis vs. alcoholics without cirrhosis and healthy subjects) [127]. In a recent study, patients with alcoholic cirrhosis had an increase in *Methanobrevibacter* and a decrease in *Catenibacterium* compared with those in patients with advanced fibrosis [110].

### 4.4. Gut Dysbiosis in AH

The background of AH mainly occurs in patients with alcoholic cirrhosis. Thus, gut dysbiosis is more altered in terms of composition and function. Llopis et al. first described that increased abundances of Enterobacteriaceae and Streptococcaceae were associated with the severity of AH [103]. Interestingly, *Atopobium* and *Clostridium leptum* were negatively associated with serum bilirubin and the degree of fibrosis. Secondary bile acids were increased with deteriorating severity of AH, an increase that was similar to the results in actively drinking patients with cirrhosis [103]. Conversely, *Atopobium* was reduced in patients with AH in the study by Ciocan et al. [124]. They demonstrated that patients with AH had increased in *Lactobacillus, Bifidobacterium, Haemophilus, Enterococcus, Streptococcus, Rothia*, and *Aggregatibacter* and decreases in *Ruminococcus, Parabacteroides, Bilophila, Odoribacter, Desulfovibrio*, and *Oscillospira* [124]. The most recent study compared the gut microbiome of patients with moderate or severe AH with that of heavy drinking subjects without hepatitis and that of nondrinking subjects. They revealed an enrichment of *Fusobacterium, Megasphaera,* and *Veillonella*. They also reported an increase in *Atopobium* in patients with AH [123]. Interestingly, patients with AH had a significant decrease in genera of the SCFA producers Lachnospiraceae and Ruminococcaceae [123,124].

## 5. Conclusions

The gut microbiome has been widely studied in human health and diseases. Its importance is increasing in the progression and deterioration of CLD, including acute decompensation of liver cirrhosis and ACLF. In particular, research on the gut microbiome in patients with ACLF, which has shown an extremely high mortality rate, is thought to be a great influence in preventing and predicting the development of ACLF, as well as in establishing short-term and long-term treatment plans through various antibiotics, probiotics, diets, and fecal microbiota transplantation. However, individualized treatment for each patient could not be suggested at present because of limitations and a lack of research according to age, pharmaceuticals, diet, race, alcohol, smoking, etiology of CLD, and comorbidities. Therefore, more extensive research is warranted—from basic studies of the composition and change of the gut microbiome according to etiology and complications of liver cirrhosis to clinical studies using the gut microbiome.

## Figures and Tables

**Table 1 ijms-22-11680-t001:** The gut dysbiosis profiles in cirrhosis.

Gut Microbial Changes	Any	Alcohol	NAFLD
Increase	**Phylum**		
	Proteobacteria [35]	Proteobacteria [36]	
	Fusobacteria [35,37,38]		
	**Family**		
	Enterobacteriacea [35,36,37,38,39,40]	Enterobactriaceae [36]	Bacteroidaceae [36]
	Enterococcaceae [36]	Prevotellaceae [35]	Porphyromonadaceae [36]
	Streptococcaceae [35,38]	Halomonadaeaace [36]	
	Pasteurellaceae [35]	Gordonibacter pamelaeae [35]	
	Veillonellaceae [35,39]		
	Virbionaceae [40]		
	Alcaligenaceae [37,40]		
	**Genus/Species**		
	*Lactobacillus* [37,38,39]	*Ruminococcus *sp.* 5_1_39BFAA* [35]	
	*Prevotella* [38]		
	*Megasphaera* [38]		
	*Campylobacter* [38]		
	*Leuconostocacea* [37]		
	*Clostridium* [38]		
	*Veillonella* [35,38,41]		
	*Streptococcus* [41]		
	*Haemophilus parainfluenzae* [38]		
	*Escherichia* [38]		
	*Shigella* [38]		
	*Salmonella* [38]		
Decrease	**Phylum**		
	Bacteroidetes [35,38]		
	Firmicutes [38]		
	**Family**		
	Clostridiales_XIV [36]	Clostridiales_XIV [36]	
	Lachnospiraceae [35,36,37,38,39,40]	Lachnospiraceae [36,40]	Veillonellaceae [36]
	Ruminococcacea [36,37,38,39]	Ruminococcaceae [36]	
	Prevotellaceae [37]		
	**Genus/Species**		
	*Alistipes* [38] *Roseburia* [38]	*Phascolarctobacterium* sp. [38]	
	*Faecalibacterium* [38]	*Bacteroides* [38]	
	*Coprococcus* [38]	*Prevotella* [38]	
	*Eubacterium* [38]	*Parabacteroides* [38]	
	*Phascolarctobacterium* [38]	*Xylaniphila* [38]	
	*Subdoligranulum* [38]	*Clostridium* [38]	
	*Bilophila* [38]	*Paraprevotella* [38]	
	*Parabacteroides* [38]	*Odoribacter splanchnicus* [38]	
	*Tannerella* [38]	*Acidaminococcus* sp. [38]	

NAFLD, nonalcoholic fatty liver disease.

**Table 2 ijms-22-11680-t002:** The gut dysbiosis profiles in ACLF.

Gut Microbial Changes	ACLF	Infection	HE	Renal Dysfunction
Increase	**Phylum**			
	Firmicutes [77]			
	Proteobacteria [77]			
	**Class**			
	Bacteroidia [77]			
	Bacilli [77]			
	Gammaproteobacteria [77]			
	**Family**			
	Enterecoccaceae [77]	Enterobacteriaceae [36,39]	Enterobacteriaceae [37]	Enterobacteriaceae [39]
	Streptococcaceae [77]	Lactobacillaceae [39]	Peptostreptococcaceae [39]	Hydrogenophilaceae [69]
	Pasteurellaceae [77]	Erysipelothricaceae [39]	Streptococcaceae- [37,39]	
	Veillonellaceae [77]	Propionibacteriaceae [39]	Staphylococcaceae [39]	
	Campylobacteriaceae [69]	Enterococcaceae [39]	Enterococcaceae [69]	
		Actinomycetales [39]	Alcaligenaceae [37]	
			Lactobacilaceae [37]	
	**Genus/Species**			
			*Enterococcus* [39]	
			*Pseudomonas* [39]	
			*Enterobacter* [39]	
Decrease	**Phylum**			
	Bateroidetes [77]	Bacteroidetes [39]		
	**Family**			
	Lanchnospiraceae [77]	Clostridiales_XIV [36]	Lachnospiraceae [77]	Lachnospiraceae [39]
	Bacteroidaceae [77]	Lachnospiraceae [36]		
	Ruminococcaceae [77]	Ruminococcaceae [36]		
	Porphyromonadaceae [77]	Veillonellaceae [36]		
		Coriobacteriaceae [36]		
		Acidaminococcaceae [39]		

ACLF, acute-on-chronic liver failure; HE, hepatic encephalopathy.

**Table 3 ijms-22-11680-t003:** Gut dysbiosis profiles in alcoholic liver disease.

Gut Microbial Changes	Mild Alcoholic Liver Disease	Alcoholic Hepatitis	Alcoholic Cirrhosis with Active Drinking
Increase	**Phylum**		
	Proteobacteria [109,111]	Fusobacteria [111]	Firmicutes [117]
	Firmicutes [111]	Actinobacteria [111]	Proteobacteria [116]
	Actinobacteria [111]	Firmicutes [111]	Enterobactericaea [127]
	Fusobacteria [111]		
	**Family**		
		Enterobactericeae [103]	Veillonellaceae [117]
		Fusobacteriaceae [123]	Peptostreptococcacae [116]
		Veillonellaceae [123]	Enterobacteriaceae [116]
	**Genus/Species**		
	*Turicella* [111]	*Turicella* [111]	*Bifidobacterium* [121]
	*Microbacterium* [111]	*Microbacterium* [111]	*Streptococcus* [121]
	*Nocardioides* [111]	*Nocardioides* [111]	*Lactobacillus* spp. [121]
	*Anaerococcus* [111]	*Anaerococcus* [111]	*Enterobacter* spp. [127]
	*Lachnospiraceae incertae sedis* [111]	*Lachnospiraceae incertae sedis* [111]	*Bacteroides* spp. [127]
	*Clostridium XI* [111]	*Clostridium XI* [111]	*Veillonella* spp. [121]
	*Klebsiella* [121]	*Curvibacte*r [111]	*Gordonibacter pamelaeae* [121]
	*Lactococcus* [121]	*Bifidobacteria* [103,124]	*Ruminococcus *sp.* 5_1_39BFAA* [121]
	*Citrobacter koseri* [121]	*Lactobacillus* [124]	
	*Lactobacillus salivarius* [121]	*Enterococcus* [124]	
	*Lactococcus lactis* subsp. *Cremoris* [121]	*Sterptococcus* [103,124]	
		*Haemophilus* [124]	
		*Atopobium* [123]	
Decrease	**Phylum**		
	Proteobacteria [109]	Bacteroidetes [111]	Bacteroidetes [117]
	Bacteroidetes [109,111]		
	**Family**		
		Lachnospiraceae [123,124]	Lachnospiraceae [116]
		Ruminococcaceae [123,124]	Ruminococcaceae [116]
			Prevotellaceae [116]
			Bacteroidaceae [117]
			Porphyromonadaceae [117]
	**Genus/Species**		
	*Prevotella* [111]	*Prevotella* [111]	*Prevotella* [121]
	*Flavobacterium* [111]	*Flavobacterium* [111]	*Paraprevotella* [121]
	*Akkermansia* [121]	*Acinetobacter* (OTU0021) [111]	*Clostridiales cluster XIV* [116]
	*Acinetobacter* (OTU0021) [111]	*Clostridium leptum* [103]	*Alistipes* [121]
	*Coprococcus* [121]	*Atopobium* [103,124]	*Parabacteroides* [121]
	*Clostridiale* [121]	*Akkermansia muciniphila* [112]	*Clostridium saccharolyticum* [121]
		*Ruminococcus* [124]	
		*Bifidobacteria* [104]	
		*Lactobacilli* [104]	*Odoribacter splanchnicus* [121]
		*Enterococci* [104]	*Phascolarctobacterium* sp. [121]

## Data Availability

Not applicable.

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
