# Peer review of "The Role of Gut Dysbiosis in Acute-on-Chronic Liver Failure"

_ijms, 2021, doi:10.3390/ijms222111680_

Round 1

Reviewer 1 Report

Gut-liver axis plays a quite important role in liver diseases. Gut bacteria and their metabolites translocate from gut to portal vein and move into liver through blood to induce inflammatory reaction. Gut-liver axis still plays the important role in acute-on-chronic liver failure (ACLF). In this review, authors described the gut dysbiosis and their function in cirrhosis, gut-brain axis in hepatic encephalopathy, progression of ACLF and alcoholic ACLF. There are a lot of information about gut microbiome changes in different liver diseases. It is a quite interesting review and will help readers understand the role of gut dysbiosis in ACLF; however, I still have a few concerns regarding the manuscript.             

  1. Gut dysbiosis is quite complex and it is difficult for readers to understand. Table would be a good protocol to present the difference of gut microbiome with different diseases, and try to reduce the word description in paper, such as 2.1, 2.2 and 3.
  2. Authors have provided 3 tables, please connect them with the word file and description.
  3. There are a lot of abbreviation in paper, please supply the whole name before the abbreviation in the first time, such as hepatic encephalopathy (HE).

Author Response

Reviewer 1

Gut-liver axis plays a quite important role in liver diseases. Gut bacteria and their metabolites translocate from gut to portal vein and move into liver through blood to induce inflammatory reaction. Gut-liver axis still plays the important role in acute-on-chronic liver failure (ACLF). In this review, authors described the gut dysbiosis and their function in cirrhosis, gut-brain axis in hepatic encephalopathy, progression of ACLF and alcoholic ACLF. There are a lot of information about gut microbiome changes in different liver diseases. It is a quite interesting review and will help readers understand the role of gut dysbiosis in ACLF; however, I still have a few concerns regarding the manuscript.             

  1. Gut dysbiosis is quite complex and it is difficult for readers to understand. Table would be a good protocol to present the difference of gut microbiome with different diseases, and try to reduce the word description in paper, such as 2.1, 2.2 and 3.

Answer: Thank you for your suggestions. According to your comment, we tried to reduce the content as using the table. Thus, the contents (page 3, line 12-16; page 3, line 19-20; page 4, line 2-5; page 4, line 25-27) were deleted, and we recommended to refer to Table 1. However, most of them might be needed to understand the role of each gut microbiome for readers.

In another study, alcohol-related cirrhotic patients had higher abundances of Enterobacteriaceae and Halomonadaceae and lower abundances of Lachnoapiraceae, Ruminococcaceae, and Clostridialies XIV. They also reported a higher abundance of Porphyromonadaceae and Bacterioidaceae and lower abundance of Veillonellaceae in nonalcoholic steatohepatitis (NASH) cirrhotic patients (page 3, line 12-16)

  • Among cirrhotic patients, there was a difference in the composition of gut microbiota between alcohol-related patients and NAFLD-related patients, which are described in Table 1.

  1. Authors have provided 3 tables, please connect them with the word file and description.

Answer: Thank you for your kind suggestions. According to your comment, we mentioned Table 1-3 in the main text.

  1. There are a lot of abbreviation in paper, please supply the whole name before the abbreviation in the first time, such as hepatic encephalopathy (HE).

Answer: Thank you for your remark. We correct that and other abbreviation. Changes have been highlighted in blue color.

Reviewer 2 Report

In this review paper, Kim et al. described changes of gut microbiome in patients with liver diseases including acute-on-chronic liver failure. Recent findings are summarized well but I have some specific comments as below.

  1. Tables 1-3 are not mentioned in the main text. Please explain the contents of Tables in the main text.
  2. Interventions on gut dysbiosis such as antibiotics are not mentioned. It would be better to include description on preclinical/clinical treatment interventions on dysbiosis.

Author Response

We are grateful for the opportunity to revise our paper (IJMS-1413034) entitled "The role of gut dysbiosis in acute-on-chronic liver failure", and the helpful comments of your reviewers. We attach a revised version, separately list our point-by-point responses. Major changes have been highlighted in blue color in the revised manuscript, tables to avoid any confusion. We feel that the comments have allowed us to improve the paper and hope you convey our gratitude to the reviewers.

Yours sincerely,

Dong Joon Kim, M.D., PhD.

Department of Internal Medicine, Hallym University College of Medicine, Chuncheon, Gangwon-do, Republic of Korea

Tel: +82-33-240-5646

E-mail: djkim@hallym.ac.kr

In this review paper, Kim et al. described changes of gut microbiome in patients with liver diseases including acute-on-chronic liver failure. Recent findings are summarized well but I have some specific comments as below.

  1. Tables 1-3 are not mentioned in the main text. Please explain the contents of Tables in the main text.

Answer: Thank you for your accurate remark. According to your comment, we mentioned Table 1-3 in the main text.

  1. Interventions on gut dysbiosis such as antibiotics are not mentioned. It would be better to include description on preclinical/clinical treatment interventions on dysbiosis.

Answer: Thank you for your suggestions. According to your comment, we added the content ‘3.3 Intervention on gut dysbiosis in ACLF’ on page 6.